# Glucocorticoid-Responsive Tissue Plasminogen Activator (tPA) and Its Inhibitor Plasminogen Activator Inhibitor-1 (PAI-1): Relevance in Stress-Related Psychiatric Disorders

**DOI:** 10.3390/ijms24054496

**Published:** 2023-02-24

**Authors:** Marie Mennesson, Jean-Michel Revest

**Affiliations:** INSERM, Neurocentre Magendie, U1215, University of Bordeaux, F-33000 Bordeaux, France

**Keywords:** stress, stress-related psychiatric disorders, anxiety, depression, compulsive and addictive behaviors, post-traumatic stress disorder (PTSD), glucocorticoid hormones (GCs), tissue plasminogen activator (tPA), plasminogen activator inhibitor-1 (PAI-1)

## Abstract

Stressful events trigger a set of complex biological responses which follow a bell-shaped pattern. Low-stress conditions have been shown to elicit beneficial effects, notably on synaptic plasticity together with an increase in cognitive processes. In contrast, overly intense stress can have deleterious behavioral effects leading to several stress-related pathologies such as anxiety, depression, substance use, obsessive-compulsive and stressor- and trauma-related disorders (e.g., post-traumatic stress disorder or PTSD in the case of traumatic events). Over a number of years, we have demonstrated that in response to stress, glucocorticoid hormones (GCs) in the hippocampus mediate a molecular shift in the balance between the expression of the tissue plasminogen activator (tPA) and its own inhibitor plasminogen activator inhibitor-1 (PAI-1) proteins. Interestingly, a shift in favor of PAI-1 was responsible for PTSD-like memory induction. In this review, after describing the biological system involving GCs, we highlight the key role of tPA/PAI-1 imbalance observed in preclinical and clinical studies associated with the emergence of stress-related pathological conditions. Thus, tPA/PAI-1 protein levels could be predictive biomarkers of the subsequent onset of stress-related disorders, and pharmacological modulation of their activity could be a potential new therapeutic approach for these debilitating conditions.

## 1. Introduction 

Although it has been well known since the 1990s that both the tissue plasminogen activator (tPA) and its inhibitor PAI-1 (type-1 plasminogen activator inhibitor) are stress-responsive proteins [1,2], containing glucocorticoid hormone (GC)-responsive elements (GRE) in their promoter sequences [3,4,5,6], their involvement in the context of stress-related psychiatric disorders has been poorly investigated in comparison with studies focusing on their role as fibrinolytic system modulators [7]. Therefore, investigating the role of the tPA/PAI-1 system in this area of research is very promising as it could bring new insights into the etiology of some psychiatric disorders triggered by stress. 

Stress-related psychiatric disorders represent a major public health problem. In 2010, they affected approximately 115 million people in Europe [8] with an estimated cost of 250 billion euros to European society [9]. Given the particularly stressful societal context (e.g., war, terrorism, and the COVID-19 pandemic), stress-related disorder incidence will rise considerably in the future. Therefore, finding new biomarkers and therapeutic targets for these diseases is of great importance.

Stress is a set of complex biological responses in the body to cope with environmental pressures [10]. For a more recent review, refer to the work of Agorastos and Chrousos who comprehensively address the neuroendrocrinology of stress concepts in terms of variations in the homeodynamic states of an organism [11]. Stressful events, whether positive or negative, can induce a higher memory trace than those with a low affective valence. These physiological responses are designed to prepare the individual to respond appropriately to a potentially dangerous situation. This includes the ability to integrate learned information about the stressor and to memorize it in order to provide an adequate and efficient response if a similar situation occurs again [12,13]. However, while at first, this coping mechanism proves to be beneficial in restoring homeostasis, prolonged activation caused by chronic stress or a single traumatic event has deleterious consequences. Prolonged activation can lead to numerous stress-related psychiatric disorders, such as anxiety, depression, compulsive and addictive behaviors, and post-traumatic stress disorder (PTSD) [14,15,16,17,18,19]. 

Alteration of cognitive performance is a feature common to most of these stress-related psychiatric diseases. Memory performance is indeed a prototypical example of the dichotomy between the beneficial and deleterious effects of stress as it follows a bell-shaped pattern. Indeed, moderate stress increases memory of the associated events, facilitating future adaptation to similar situations [13,19,20]. Conversely, chronic stress or traumatic events can impair cognitive performance (e.g., memory consolidation), leading to pathological states such as those observed in stress-related disorders [13,15,21]. While from a phenomenological point of view, the effects of environmental stimuli on memory performance have been well demonstrated, the cellular and molecular mechanisms underlying these effects are still poorly understood. 

One of the major biological systems involved in responses to stress is the activation of the hypothalamic–pituitary–adrenal (HPA) axis via the secretion of its end biological product: GCs. With a biphasic bell-shaped response, GCs are one of the factors that contribute to the transition from the beneficial to deleterious effects of stress on memory performance [22,23,24]. Over the past 20 years, we have studied this transition and have shown that GCs, through glucocorticoid receptor (GR) activation in the hippocampus (HPC), one of the major brain structures involved in contextual memory processing, regulate the expression of tPA and PAI-1 proteins. In particular, we have demonstrated that a shift in the tPA/PAI-1 balance in favor of PAI-1 is responsible for the transition from the beneficial to deleterious effects of stress on cognitive performance, modeling significant aspects of PTSD [25,26,27,28,29]. 

This review aims to address the major role played by tPA/PAI-1 balance modulated by GR activation in response to GC release in the context of stress-related psychiatric disorders. Our hypothesis is that this balance is crucial in the mediation of the pathophysiological effects of stress on brain functions, notably cognitive performance. Therefore, deregulations in this balance could explain the negative effects of intense stress on cognitive performance, such as those observed in stress-related psychiatric disorders. Here, after introducing the GC-responsive tPA/PAI-1 system, we review findings which could support this hypothesis. While data from preclinical and clinical literature related to stress-related psychiatric disorders sometimes differ, most studies agree on the fact that the tPA/PAI-1 system is deregulated in these diseases.

## 2. Glucocorticoid Hormones (GC), Their Receptors, and Mechanism of Action 

GCs are the main stress steroid hormones and can easily cross the blood–brain barrier (BBB) because of their lipophilic structure. They are cortisol in humans and corticosterone in rodents, which differ structurally through one hydroxy group (Figure 1). Termed tonic or phasic, GC secretion is modulated by the circadian rhythm and in response to stress. While stable basal levels are maintained by tonic secretion (i.e., 1 µg/100 mL in plasma), phasic secretion is engaged at the beginning of the active phase and during environmental stimulation, resulting in higher levels of circulating GCs (i.e., 10 µg/100 mL) [30]. Furthermore, their secretion in response to stress can increase basal levels by a factor of over 40 [31].

The adrenal glands are responsible for GC secretion and are the final step of the HPA axis [31]. This axis is one of the many systems involved in the body’s response to environmental conditions, and there is a consensus that GC secretion constitutes a crucial response to protect the body against environmental threats [32]. HPA axis activity is controlled by numerous central afferences, including the suprachiasmatic nucleus of the hypothalamus (SCN) for circadian rhythm modulation and the HPC, the amygdala (AMY), and the prefrontal cortex (PFC) for stress responses. First, in response to a neuroendocrine signal from the hypothalamus (HT) paraventricular nucleus (i.e., corticotropin-releasing factor [CRF] and/or vasopressin [AVP]), the anterior pituitary gland secrets adrenocorticotropic hormone (ACTH) causing a discharge of GCs into the blood stream through the adrenal glands. Circulating GCs are 90% bound to corticosteroid-binding globulin (CBG). The rest (10%) are albumin bound or free, able to cross the BBB, thus corresponding to the active fraction of the central nervous system (CNS) [33,34]. Finally, GCs play an inhibitory feedback control on their own secretion by modulating every stage of the HPA axis (Figure 2). GCs are involved in numerous physiological regulations at the CNS level, such as neuronal and behavioral function, but also at the periphery, including modulation of cardiovascular, immune, inflammatory, respiratory, and metabolic systems [35].

At the cellular level, 11β-hydroxysteroid dehydrogenase (11β-HSD) type I and II enzymes regulate the availability of intracellular GCs and thus their ability to activate their receptors, which are mostly localized in the intracellular shaft. In fact, GCs are able to bind two cytoplasmic receptors: the mineralocorticoid and glucocorticoid receptors (MRs and GRs), which are transcription factors that cooperate through genomic but also rapid nongenomic mechanisms to coordinate acute and chronic stress responses associated with peripheral neuroendocrine and central social, cognitive, and emotional processes [36,37]. However, GCs having a stronger affinity for MRs than GRs results in MRs being saturated at the hormone’s basal levels. Therefore, GRs are significantly activated only when hormone levels are high, such as those observed during circadian peak and environmental challenges. For this reason, GRs are considered the main stress receptor. They are expressed throughout peripheral tissues such as the lung, heart, liver, kidney, etc., but also in many cerebral structures [38]. Indeed, GRs are found in numerous brain regions and different neural (neurons, glial cells) and non-neural (endothelial) cell types. Their expression level is particularly high in the HPC, notably in the dorsal area [39,40]. This limbic region, which encodes contextual and spatial memory, is strongly modulated by emotions and is thus involved in the etiology of many psychiatric disorders such as anxiety, depression, and PTSD [41]. In the absence of GCs, GRs are found in the cytoplasm as a conformationally inactive protein complex with chaperones immunophilin FK506-binding proteins (FKBP) and heat shock proteins (Hsp). After binding to GCs, they are activated by phosphorylation and translocated into the nucleus where they modulate, as homo- or heterodimers, gene transcription through three known mechanisms: (1) remodeling chromatin via the recruitment of enzymes affecting histone post-translational modification and DNA methylation [42,43], (2) direct DNA binding on glucocorticoid-responsive elements (GREs) or negative GRE (nGRE) sequences present in the promoter region of target genes [36,44,45], and (3) interaction with other transcription factors, such as Jun, CREB, Fos-Jun, and NfκB [36,44,45] (Figure 3). The various modulations produced by translocated GRs lead to the downstream molecular and behavioral effects of phasic GC secretion.

Although it is generally agreed that the majority of the pathophysiological effects of stress are mediated through GR activation, it is important to remember that they are expressed ubiquitously by most neural and non-neural cells. Therefore, we have little information about the cellular and molecular targets of GRs that regulate their pathophysiological effects on cognition and adaptive behavior. The understanding of GC cellular and/or tissue-specific substrates is of key importance to better appreciate the mechanism of stress-induced pathologies and to suggest new diagnostic and therapeutic strategies.

## 3. GMES Signaling Cascade in the HPC and Bell-Shaped Pattern of GC Effects on Cognitive Processes

A bell-shaped curve representation is commonly used to depict the beneficial versus the deleterious effects of stress. Notably, numerous studies have demonstrated a bell-shaped effect triggered by GCs on cognitive performance [22,23,24,46,47]. Indeed, GC increases induced by low-to-moderate stress are known to improve memory encoding [19,48,49]. Whereas, chronic or intense acute stress is associated with memory decline [15,50,51]. For example, in the case of PTSD, patients show declarative memory deficits following a traumatic event. Notably, they present a higher amplitude of secreted cortisol when facing stimuli present during the traumatic event, reflecting hyper-reactivity of their HPA axis [52]. 

Over the past 20 years, our laboratory has been investigating the molecular effectors involved in this bell-shaped curve through the pivotal role of GR mediating GC cellular effects. Using a wide variety of approaches, we identified a signaling cascade within the HPC which could account for the beneficial effect of stress on memory encoding [25,26,27,28]. This cascade was named GMES after identifying the main effectors: **G**R-tPA-BDNF-TrkB-Erk1/2**^M^**^APK^-**E**gr-1-**S**ynapsin-Ia/Ib. Briefly, during moderate stress, GC-activated GRs stimulate both the expression of the immature form of the brain-derived neurotrophic factor (pro-BDNF) and tPA [27] in the HPC through indirect or direct gene expression regulation (e.g., GREs present in the tPA promoter region) [5,6]. Then, the tPA increase triggers the cleavage of plasminogen to plasmin, which further cleaves the pro-BDNF to the mature form of BDNF (m-BDNF). m-BDNF, by activating the tyrosine kinase receptor B (TrkB), phosphorylates Erk1/2^MAPK^ intracellular kinases which, by increasing the expression of the downstream transcription factor Egr-1 [25,28], finally enhance the level and activity of promnesic proteins such as synapsin Ia/Ib [26] (Figure 4). Notably, these studies highlighted the fundamental role of GC-induced tPA in memory encoding.

Our latest work was focused on the deleterious effect of GCs on cognitive performance. Our hypothesis was that the deregulation of the GMES cascade in the HPC could be responsible for a shift from normal to pathological memory such as that observed in PTSD patients. Searching for effectors able to downregulate the whole GMES cascade, we targeted PAI-1, the role of which is to inhibit tPA. Interestingly PAI-1 is also expressed in the HPC and contains GREs in its promoter region [3]. Indeed, using a mouse model displaying the paradoxical memory features of PTSD (i.e., contextual amnesia and emotional hypermnesia) [15], we showed that the deregulation of the GMES signaling cascade via an increase in PAI-1 induced by intense or traumatic stress was responsible for the formation of pathological memory, modeling important aspects of PTSD [29] (Figure 4). The dynamic modulation of the tPA/PAI-1 balance has provided insight regarding the paradoxical duality of the bell-shaped behavioral effects of GCs being able to regulate both tPA and its inhibitor PAI-1. Our data thus help to explain how GCs can have opposite functional and behavioral effects: stimulating for moderate stress and deleterious for intense stress such as chronic or traumatic stress. In addition to GMES cascade inhibition, an increase in PAI-1 levels is responsible for pro-BDNF accumulation by blocking its conversion into m-BDNF by the tPA/plasmin system. Long considered inactive, pro-BDNF forms are actually able to form a ternary complex with p75^NTR^ and sortilin pro-apoptotic receptors to induce neuronal cell death by apoptosis [53] and decrease synaptic plasticity (LTD induction) [54], in contrast to BDNF/TrkB signaling, which is known to promote long-term potentiation (LTP) [55]. Thus, pro-BDNF accumulation could also be involved in the physiopathology of the impaired cognitive function observed in this mouse model (Figure 4). 

## 4. The tPA/PAI-1 System in Health and Diseases

### 4.1. tPA and Its Inhibitor PAI-1: Structure, Function, and Expression Pattern

tPA, encoded by the *PLAT* gene, is a 70 kDa enzymatically active serine protease with a short half-life (2–6 min), known principally for its role in the vascular system [56], in particular for blood clot lysis (fibrinolysis). tPA is mainly produced and released by endothelial cells in the blood circulation, which mediates the cleavage of plasminogen to plasmin, later degrading fibrin to fibrin degradation products (FDP) (Figure 5). tPA has demonstrated its therapeutic efficacy as a thrombolytic agent in the elimination of blood clots, notably in the treatment of myocardial infarction and ischemic stroke [57]. Interestingly, tPA is also present in the CNS where it plays a key role as a mediator of pro-BDNF to m-BDNF conversion (Figure 4). In the brain, tPA is expressed in the HT, AMY, bed nucleus of stria terminalis (BNST) [58], visual cortex [59], SCN [60], and HPC [61,62]. Notably, in the HPC, tPA is localized in glutamatergic neurons of the dentate gyrus (DG) and of the Cornu Ammonis 1, 2, and 3 (CA1, CA2 and CA3) [63,64,65]. In addition to its neuronal expression, tPA mRNA is also found in astrocytes [66], oligodendrocytes [67], and endothelial cells in the CNS [65].

tPA activity is closely regulated by specific inhibitors belonging to the serine proteinase inhibitor (serpins) superfamily [68] whose role, among others, is to inhibit the formation of plasmin from plasminogen. Although there are many serpins present in all biological species [69], in mammals, there are essentially four plasminogen activator inhibitors: PAI-1, PAI-2, neuroserpin, and nexin-1 protease. However, PAI-1 is the main inhibitor of tPA [70] and has also been most closely studied because of its role in the cardiovascular system. PAI-1, encoded by the *SERPIN1* gene, is a 50 kDa protein directly secreted in its active form and has a half-life of 2 h [71]. Binding to vitronectin maintains PAI-1 in its active conformation, allowing for the prolonged inhibition of plasmin formation [72]. In the blood stream, PAI-1 is secreted by endothelial cells and regulates the fibrinolysis process via inhibition of plasminogen to plasmin conversion by tPA (Figure 5). As tPA, PAI-1 is also expressed in the CNS, but its basal expression is relatively low compared to the vascular system [73]. PAI-1 is produced by neurons [74,75,76] and astrocytes [76,77,78]. In basal conditions, a low level of expression is detected in the cortex, HT, AMY [76], and SCN [60]. After stimulation by kainic acid, PAI-1 is found in the DG and CA3 regions of the HPC [64] where its spatiotemporal pattern matches that of GRs [79]. PAI-1 is also detected in the nucleus accumbens (NAc) of mice in response to morphine [80]. Finally, in response to threats such cerebral ischemia, PAI-1 expression is strongly increased in the lesioned cortex, while the level of other tPA inhibitors such as neuroserpin and protease nexin-1 are unaffected [77]. In addition, in order to contain the GRE consensus sequence in its promoter region as for the tPA encoding gene [3,4], the PAI-1 encoding gene contains a canonical E-Box motif, also sensitive to GC-activated GRs, which has been shown to trigger the rhythmic secretion of PAI-1 [81].

### 4.2. The tPA/PAI-1 System Outside the CNS

Due to its primary function and expression profile, the role of the tPA/PAI-1 duo was initially and extensively studied outside the CNS. The tPA/PAI-1 balance has in fact been widely addressed in mammals, notably in mice using different transgenic models, but also in humans through blood samples or post-mortem tissue analysis. A deregulation in the balance in favor of PAI-1 generally coincides with the development of pathological processes. These can be reversed by inhibiting PAI-1 activity using small-molecule inhibitors [82,83,84,85] suggesting that the tPA/PAI-1 system is a potential therapeutic target for many pathological conditions, such as cardiovascular diseases, metabolic disorders, fibrosis, cancer, and inflammatory and infectious diseases (Table 1). Since the focus of this review is on stress-related psychiatric disorders, we will not develop this section further. For a non-exhaustive list of review articles on this topic see Table 1.

### 4.3. Preclinical Studies Highlighting the tPA/PAI-1 System in the CNS

Since tPA and PAI-1 are both expressed in the CNS, notably in brain areas related to learning and memory, their involvement in memory and underlying synaptic mechanisms has been investigated over the past 30 years. Cognitive processes such as learning and memory have been strongly linked to a cellular process, classically called long-term synaptic plasticity, which corresponds to a persistent modification of synaptic strength [99]. Numerous studies support a major role of tPA in synaptic plasticity [100]. In the HPC, tPA mRNA was over-expressed less than 30 min following LTP induction, which could account for important cellular rearrangements [101,102]. Indeed, perfusion of rat hippocampal slices with tPA and transgenic mice over-expressing tPA (Thy^tPA^) both showed an increased LTP in the HPC, processes which can be reversed with specific tPA inhibitors [102,103]. Furthermore, mice knock-out (KO^tPA^) for the tPA encoding gene displayed a reduced LTP [104]. Accordingly, Thy^tPA^ mice showed better memorization in the Morris water maze test than their wild-type littermates [103], and an increase in the tPA protein level was observed in the cerebellum following motor learning [105]. KO^tPA^ were deficient in step-down inhibitory avoidance learning, a hippocampus-dependent task [106]. Moreover, pro-BDNF cleavage by the tPA/plasmin system was essential for late LTP in the mouse HPC [55]. In addition, based on our past published results, tPA is clearly able to rapidly activate the intracellular signaling cascade involved in memory consolidation such as the Erk1/2^MAPK^ pathway [25,26,27]. According to its role as a tPA inhibitor, PAI-1 was shown to inhibit synaptic plasticity in hippocampal primary neuronal cultures [107]. Altogether, these findings are consistent with a promnesic role of tPA. 

Several studies have investigated the pathophysiological role of tPA on neuronal survival at the cellular level. While some of these studies reported a neuroprotective effect of tPA [108,109,110], tPA-dependent neurotoxic effects have also been demonstrated. For example, KO^tPA^ mice were resistant to neuronal degeneration induced by excitotoxin [111] and presented less ischemic damage following a stroke [112]. These neurotoxic effects of tPA might be linked to its role as a modulator of NMDA receptor (NMDAR) signaling and be independent of the plasminogen to plasmin conversion modulation. Indeed, in vitro tPA delivery was shown to induce cortical neuron apoptosis through Ca^2+^ accumulation mediated by NMDAR [113]. Accordingly, PAI-1 and anti-tPA antibodies were able to block neuronal death induced by kainic acid [114] most likely via the inhibition of tPA-dependent cytotoxicity.

Finally, numerous studies have also reported the major involvement of the tPA/PAI-1 system in response to stress. Interestingly, as previously mentioned, both the tPA and PAI-1 encoding genes contain GRE sequences in their promoter regions, indicating that GCs play a central role in regulating their expression when mediating the behavioral consequences of stress. Concerning tPA, some studies have shown that its activity is potentiated by intense or chronic stress. Indeed, a 30 min restraint stress or an intraventricular injection of CRF stimulated its expression in the AMY together with increased anxiety-like behavior. Interestingly, KO^tPA^ mice did not display an increased level of anxiety when confronted with a similar treatment [115,116]. In addition, WT mice that went through chronic restraint stress had a loss of dendritic spines in the medial AMY and the dorsal HPC, while KO^tPA^ mice presented a reduced effect from this procedure [117,118]. KO^tPA^ mice were protected from impaired spatial hippocampo-dependent memory following chronic stress [117]. Altogether, these studies have provided evidence that, in response to intense or chronic stress, tPA increases might have deleterious effects on cognitive function and synaptic plasticity in the AMY and HPC. However, in these studies, the expression of PAI-1 was not investigated and could likely be upregulated along with tPA. 

We actually observed opposite effects of tPA in the HPC following moderate acute stress. When we applied a 30 min acute restraint stress, we were able to correlate an increase in tPA levels with stress-related memory potentiation using a fear conditioning paradigm together with elevated levels of promnesic p-Erk1/2^MAPK^ [27], whereas, after 1 h and 3 h of restraint stress, we found an increase in PAI-1 levels instead, which was correlated with a decrease in p-Erk1/2^MAPK^ levels [29]. In addition, a similar observation in mice with a PTSD-like memory profile was observed [29]. We therefore concluded that an imbalance of the tPA/PAI-1 ratio in favor of PAI-1 was a consequence of intense acute stress. Accordingly, inhibiting PAI-1 directly in the HPC by an injection of tiplaxtinin (PAI-039) prevented memory impairment triggered by toxic levels of GCs [29]. Few studies have shown that restraint stress in mice increases the PAI-1 expression level in many peripheral tissues, such as in adipose and liver tissues [2], but also in the brain including in the AMY [115] and the HPC [29]. Interestingly, in human PAI-1 levels are increased in the elderly [2,119] together with higher levels of GC associated with cognitive decline [120,121]. In addition, in an adult healthy cohort, plasmatic levels of PAI-1 were positively correlated with their stress states [122]. Similar results were observed in a cohort of caregivers at a psychiatric hospital responsible for patients suffering from dementia [123]. 

Overall, these studies revealed that tPA and PAI-1 proteins play an important role in synaptic plasticity, cognition, and memory but also in anxiety-related behaviors in response to stress. In the HPC, the tPA/PAI-1 balance is crucial for mediating the pathophysiological effects of stress on cognitive performance, while in the AMY, tPA was shown to have a major role in anxiety-related behaviors. Their roles might therefore differ depending on the brain regions, cellular subtypes (neural vs. glial) in question, and likely supporting different molecular mechanisms. Understanding the selective cellular distribution of the tPA/PAI-1 modulatory system in response to stress in these brain regions is of importance for a better comprehension of their involvement in stress-related disorders.

### 4.4. Clinical Studies Highlighting the Involvement of tPA/PAI-1 System in Stress-Related Psychiatric Disorders

A review of the potential involvement of the tPA/PAI-1 system in stress-related psychiatric disorders as they are described by the Diagnostic and Statistical Manual of Mental Disorders (DSM-5) and the International Classification of Diseases, 11th Revision (ICD-11) was performed. According to the literature, the tPA/PAI-1 system could, to a variable extent, play an important role in the occurrence of stress-related psychiatric disorders. These are characterized by major clinical alterations in the individual’s cognitive state, emotion regulation, or behavior. We will focus on the literature regarding the tPA/PAI-1 system in particular on (1) anxiety disorders, (2) depressive disorders, (3) substance use disorders (SUD), (4) obsessive-compulsive spectrum disorders, and (5) stressor- and trauma-related disorders. From the study conducted by Wittchen and colleagues, when calculating the percentage prevalence of these five disorders as part of a whole, anxiety and depression represent the most commonly observed stress-related disorders, while obsessive-compulsive spectrum disorders are the least represented class (Figure 6) [8]. It is important to mention that, although some of these disorders have been shown to elicit distinct symptoms, they share some significant overlapping. Indeed, common symptoms are observed in anxiety, major depressive disorder (MDD), and PTSD. For example, difficulties with concentration, fatigue, irritability, memory and sleep troubles, negative thoughts, extreme worry, etc.

In 2010, around 24% of Europeans were diagnosed with these previously listed disorders, representing an estimated cost of €250 billion for European society [8,9]. In 2019, the World Health Organization (WHO) estimated that 1 in 8 people worldwide suffer from a mental disorder, with anxiety and depression being the most common. Furthermore, since 2020, the current societal context (e.g., war, terrorist attacks, post-COVID-19 respiratory complications) suggests that there will be a rise in mental health disorders, notably PTSD [124,125]. For example, as a direct consequence of the COVID-19 pandemic, first estimations indicate an increase of 26% and 28% for anxiety and MDD, respectively, in just one year [126] and up to a 35% prevalence for PTSD in subjects who have encountered post-COVID-19 respiratory complications [127]. Since tPA and PAI-1 circulate in body fluids and are upregulated upon stress, they represent promising biomarkers and potential drug targets for stress-related disorders. 

#### 4.4.1. Anxiety Disorders

Anxiety disorders are the most common stress-related disorders, their prevalence being estimated at 11% in 2010 in Europe. This estimation includes generalized anxiety disorder (GAD, 2%), panic disorder (PD, 1.2%), social phobias (SP, 2%), agoraphobia (1.2%), and specific phobias (4.9%) [8]. Anxiety disorders can be defined by subjective reports of tension or excess of worry, usually co-occurring with physiological somatic symptoms (e.g., elevated heart rate or blood pressure) [128]. Although several preclinical studies have highlighted the importance of tPA/PAI-1 in anxiety-like behavior [56,112,113], only a few clinical studies have measured the tPA/PAI-1 balance in anxiety disorders and are far from being conclusive in demonstrating an association. Indeed, in one clinical study, significant tPA/PAI-1 activation was observed in a group of anxiety subjects with PD (with agoraphobia) or SP [129] (Table 2). However, in another study on a PD cohort, there were no significant differences in either tPA or PAI-1 blood levels [130] (Table 2). 

#### 4.4.2. Depressive Disorders

Depression is a very common pathologic condition worldwide, with an estimated prevalence of 5% in adults according to the WHO in 2019. The main symptoms of depression are feelings of sadness, fatigue, and an aversion to activity, body weight variation, and loss of motivation (e.g., anhedonia) [128]. In addition, depression is highly comorbid with anxiety [131], PTSD [132], and SUD [133]. At the preclinical stage, only a few studies on tPA/PAI-1 and depression showed diverging results. While KO^PAI−1^ mice presented a high score for depression-like behavior, KO^tPA^ mice did not show a clear phenotype using the same paradigms, suggesting that PAI-1 deficiency predisposes depression [134]. Conversely, using the chronic unpredictable mild stress (CUMS) procedure in rats as a model of depression, Jiang and collaborators showed an increase in PAI-1 levels in several brain subregions, cerebrospinal fluid, and the serum of depressed rats [135]. In addition, pro-BDNF to m-BDNF conversion by tPA was linked to improved depressive-like behavior in rats treated with ketamine or electroacupuncture [136,137]. These latest preclinical studies are more consistent with several clinical studies which have also investigated the role of the tPA/PAI-1 system in depression. Notably, elevated blood levels of PAI-1 have been associated with the pathogenesis of depression in MDD cohorts mainly composed of women [135,138] in a first-episode depression cohort [139] and in depressed men [140] (Table 2). However, Chen and colleagues observed no difference in PAI-1 levels in association with MDD [130]. Altogether, these studies did not measure or report changes in tPA levels (Table 2). Interestingly, Jiang and colleagues also reported reduced levels of circulating BDNF in addition to increased levels of PAI-1 in subjects with MDD [135] (Table 2). Finally, genetic polymorphisms in the *SERPIN1* gene were found to be associated with depression, suggesting that genetic variants in PAI-1 may play a role in the pathogenesis of MDD [141]. Overall, these clinical data support tPA/PAI-1 deregulation in favor of PAI-1 in association with depressive disorders. 

#### 4.4.3. Substance Use Disorders (SUD)

SUD is a compulsive use of drugs that ultimately affects brain connectivity and behavior, resulting, in turn, in the inability to control consumption of legal or illegal drugs, alcohol, and medication despite the harmful consequences [128]. According to Wittchen and colleagues, SUD affected 4% of people in Europe in 2010, with the majority being due to alcohol dependence (3.4%) and the rest an addiction to opioids (0.3%) and cannabis (0.3%). In 2019, the WHO estimated that 35 million people worldwide were suffering from SUD. Comorbidity between SUD and stress-related disorders was demonstrated a long time ago with notably seminal studies showing that alterations in GC secretion induced by chronic stress are involved in drug abuse [18,142]. Only a few preclinical and clinical studies have investigated the involvement of the tPA/PAI-1 system in SUD, mostly concerning its ability to modulate BDNF signaling in response to withdrawal from morphine [143], cocaine [144], and alcohol [145]. The data showed that chronic morphine use increases tPA expression, while withdrawal reduces it [143,146]. Zhou and collaborators demonstrated that chronic “binge” cocaine consumption dramatically decreased tPA activity by 50% in the central and medial AMY and induced a two-fold increase in amygdalar PAI-1 in comparison to saline-injected mice [144]. The clinical study driven by Delahousse and collaborators clearly demonstrated an increase in overall fibrinolytic activity after alcohol withdrawal, which was mainly due to a decrease in PAI-1 levels [145] (Table 2). All these studies suggested that the tPA/PAI-1 system plays a role in the behavioral effects of SUD and on withdrawal.

#### 4.4.4. Obsessive-Compulsive Spectrum Disorders

In the previous version of the DSM (i.e., DSM-4), these disorders were included under the anxiety disorder class and had a prevalence of 0.7% in Europe in 2010 [8]. They are characterized by repetitive thoughts, distressing emotions, and compulsive behaviors. This class is composed of obsessive-compulsive disorders (OCD), body dysmorphic disorder, hoarding disorder, trichotillomania, and excoriation disorder. OCD was the subject of most studies of these disorders, and the others were placed in this group based on symptom similarities. Unfortunately, there are no preclinical, even though mouse models exist [147,148], or clinical studies investigating the tPA/PAI-1 system in these disorders. However, a few recent studies have shown BDNF deregulations in OCD. Indeed, induced OCD stereotypic behaviors in mice were completely reversed by BDNF infusion [149]. In correlation with this preclinical finding, a computational study reassembling results from 9 clinical studies on OCD patients demonstrated reduced levels of circulating BDNF [150]. This could be explained by an inhibition of plasminogen to plasmin conversion by tPA [55] through increased levels of PAI-1 [29]. New studies are needed to further evaluate the involvement of the tPA/PAI-1 system in the etiology of obsessive-compulsive spectrum disorders.

#### 4.4.5. Stressor- and Trauma-Related Disorders

Stressor- and trauma-related disorders are a new class of disorders emerging in the DMS-5, which were previously referenced under anxiety disorder [128]. The most common disorder from this class is PTSD, but it also comprises reactive attachment disorder, disinhibited social engagement, acute stress disorder, and adjustment disorder. PTSD is a severe stress-related disorder with an estimated lifetime prevalence of about 8% in the general population, affecting 7.7 million people in Europe which represents 8.4 billion EUR in costs [8,9]. PTSD occurs in approximately 10 to 20% of subjects who have experienced an intense and traumatic stress situation in which their physical and/or psychological health has been threatened (e.g., terrorist attack, sexual abuse, military fights, post-COVID-19 respiratory complications). One of the cardinal features of this psychiatric disorder is a qualitative and paradoxical alteration of memory. This memory alteration is characterized by the coexistence of hypermnesia for certain salient elements of the event, although not predictive of the trauma (e.g., odors, ambient noises, objects, etc.), and declarative amnesia for the context in which the trauma occurred. Having focused their attention on these salient stimuli as opposed to the context of the trauma, PTSD subjects subsequently exhibit maladaptive fear responses to these stimuli in nontraumatic contexts [128]. CNS-related complications of PTSD are frequent (e.g., depression, early dementia, etc.), which can lead to chronic PTSD.

As already stated in this review, our most recent published data is the first preclinical study showing that tPA/PAI-1 imbalance is involved in the formation of PTSD-like memory in mice [29]. Thus, variation in tPA and PAI-1 blood concentrations may represent biomarkers of high predictive value for the diagnosis of PTSD. Furthermore, it is known that PTSD patients show comorbidities with peripheral diseases such as cardiovascular pathophysiology, notably atherothrombosis [151], for which high levels of PAI-1 is a known risk factor [152]. Because not all subjects who have experienced a major traumatic event develop PTSD, the identification of circulating biomarkers in body fluids is crucial for the pre/post symptomatic diagnosis of this pathology for which no clear biomarker has yet been identified. Few clinical studies have investigated the relationship between PTSD and peripheral tPA/PAI-1. Although the results of these studies showed some discrepancies, overall, they point to a deregulation of the tPA/PAI-1 system (Table 2). Briefly, two studies reported higher levels of circulating tPA in PTSD patients [153,154]. Of these two studies, only Maguire and colleagues tested PAI-1 levels and observed no difference [154], whereas a study from Farr and colleagues did not measure tPA levels but observed higher levels of PAI-1 in PTSD patients compared to healthy controls [155]. In addition, Aksu and colleagues reported reduced levels of circulating BDNF coherent with an over-expression of PAI-1 [153]. Furthermore, although tPA/PAI-1 imbalance was not investigated in a study from Stratta and colleagues, clinical correlates were demonstrated between BDNF expression and PTSD in subjects experiencing a traumatic event (i.e., earthquake-type natural disaster), showing a lower BDNF level for PTSD subjects than subjects with partial PTSD and controls [156]. Accordingly, soldiers from Afghanistan with PTSD together with depression symptoms presented lower plasmatic BDNF levels compared to non-PTSD soldiers [157]. It is important to highlight that discrepancies observed in comparing these studies are probably due to experimental limitations in humans such as numbers, heterogeneity of the subjects included, traumatic contexts, and diverse diagnostic methods [158]. However, overall, these studies point in the direction of a deregulation of the tPA/PAI-1 system, which could cause an increased inhibition of pro-BDNF to m-BDNF conversion.

To summarize this section, both preclinical and clinical studies have provided clear evidence of tPA/PAI-1 system imbalance in stress-related disorders. These deregulations seem to go toward an over-expression of both tPA and PAI-1, the latter by inhibiting the activity of tPA, (1) decreasing the availability of m-BDNF (2) which causes accumulation of pro-BDNF (Figure 4). Deregulation of BDNF is well known in psychiatric disorders [159], but the molecular mechanisms involved are still poorly understood. Based on our findings and others from the literature [25,26,28,29,55,160], signaling cascade deregulations linked to pro-BDNF/m-BDNF balance are clearly involved in the cognitive impairment observed in stress-related disorders. Therefore, it is intuitive to think that, *via* a “domino” effect, upstream deregulation of the tPA/PAI-1 system also plays an important role in stress-related psychiatric disorders. However, due to the limitations of clinical studies, it has been difficult to clearly demonstrate such deregulations. Differences in patients’ sex, age, country of origin, treatment, etc., account for the strongest limitation in comparing data from articles already present in the literature (Table 2). 

**Table 2 ijms-24-04496-t002:** Summary of findings from the literature concerning the tPA/PAI-1 system in the blood of human cohorts with stress-related psychiatric disorders. nt = not tested, nm = not mentioned, n = sample size, W = women, M = men, MDD = major depressive disorder, PD = panic disorder, SP = social phobia, SUD = substance use disorder, and PTSD = post-traumatic stress-disorder.

Disorders	tPA	PAI-1	BDNF	Cohorts Tested	n (n/Sex)	Age (Years)	Population	References
**Anxiety**	**↗**	**↗**	nt	PD (with agoraphobia) & SP	29 (21 W/8 M)	38.1 ± 10.5	Germany	[129] Geiser et al., 2008
	→	→	→	PD	30 (15 W/15 M)	35.5 ± 12.38	Chinese	[130] Chen et al., 2017
**Depression**	nt	**↗**	nt	MDD	45 (45 W)	37 ± 6.8	American	[138] Eskandari et al., 2005
	→	**↗**	nt	Depressive men	49 (49 M)	40–65	French	[140] Lahlou-Laforet et al., 2006
	nt	**↗**	**↘**	MDD	17 (14 W/3 M)	48 ± 12	Chinese	[135] Jiang et al., 2017
	→	→	→	MDD	30 (25 W/5 M)	41.60 ± 12.43	Chinese	[130] Chen et al., 2017
	→	**↗**	nt	First-episode depression	44 (21 W/23 M)	25.89 ± 1.116	Chinese	[139] Han et al., 2019
**SUD**	**↗**	**↗**	nt	Alcoholic before vs. after withdrawal	10 (10 M)	27–55	French	[145] Delahousse et al., 2001
**PTSD**	nt	**↗**	nt	PTSD (high severity score)	40 (27 W/13 M)	44.9 ± 3.5	American	[155] Farr et al., 2015
	**↗**	nt	**↘**	PTSD sexually abused child	45 (44 W/45 M)	14.7 ± 2.5	Turkish	[153] Aksu et al., 2018
	**↗**	→	→	PTSD	20 (11 W/9 M)	41.5 ± 11	Ireland	[154] Maguire et al., 2021

## 5. Conclusions

Research over the years has confirmed the pleiotropic effects played by the tPA/PAI-1 system in the CNS. In particular, the literature has provided clear evidence of tPA/PAI-1 system imbalance in stress-related disorders likely supported through the central role of GCs in mediating the behavioral consequences of stress. Notably, we have provided a first-molecular explanation for the phenomenological evidence of the bell-shaped effects of GCs on stress-related memory events. This mechanism might explain the functional duality of GCs through the transcriptional activity of the GR [25,26,27,29]. In particular how GCs, by temporally regulating both tPA and then PAI-1, can elicit opposite functional and behavioral effects depending on the stress magnitude, being stimulatory in the case of moderate stress and deleterious for chronic or traumatic stress [49]. Such a mechanism could explain the etiology of many stress-related disorders such as PTSD, anxio-depressive disorders, and other pathological conditions with overlapping symptoms. Therefore, overall, this evidence is in agreement with the potential use of tPA and PAI-1 proteins as diagnostic biomarkers and therapeutic targets for stress-related pathological conditions.

## Figures and Tables

**Figure 1 ijms-24-04496-f001:**
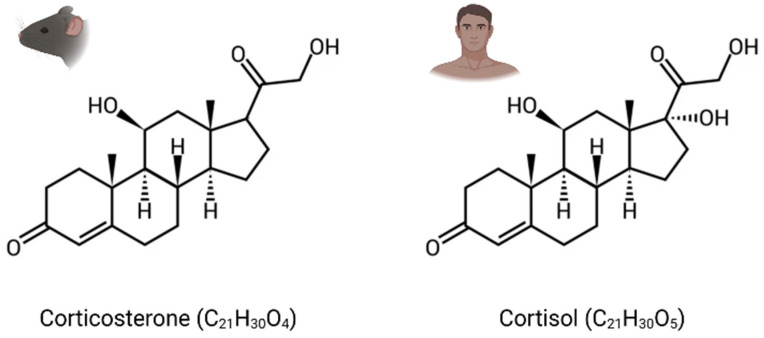
Chemical structure of glucocorticoid hormones (GCs). Corticosterone: corticoid present in rodents. Cortisol: corticoid present in humans.

**Figure 2 ijms-24-04496-f002:**
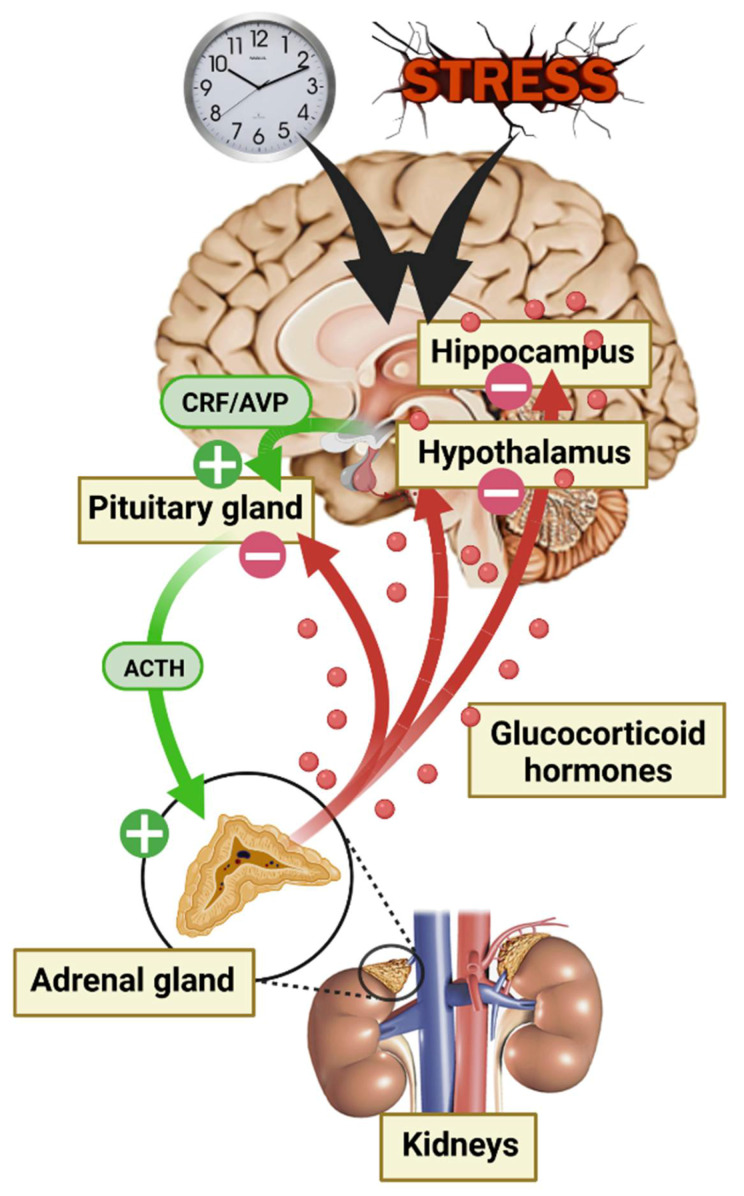
Diagram of hypothalamic–pituitary–adrenal (HPA) axis activation during the circadian cycle and in response to stress.

**Figure 3 ijms-24-04496-f003:**
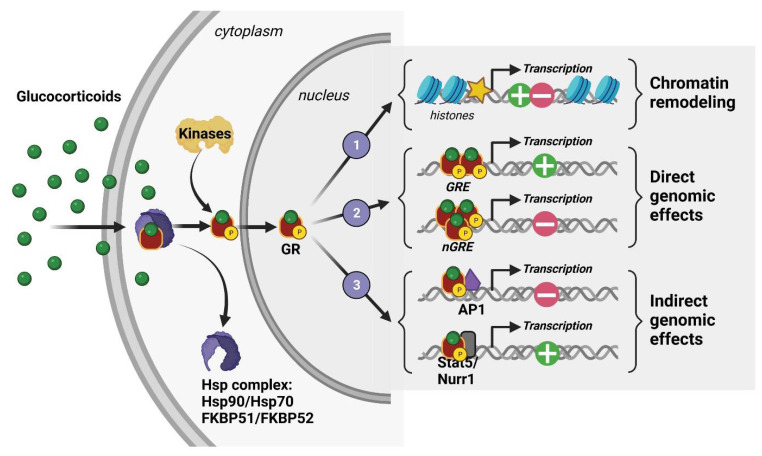
Molecular mechanisms of action of a glucocorticoid receptor (GR).

**Figure 4 ijms-24-04496-f004:**
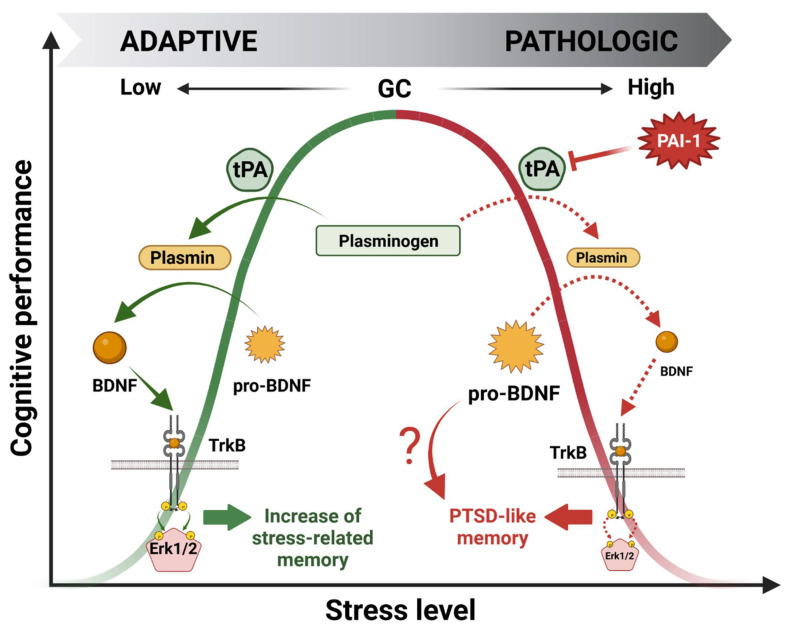
Molecular mechanisms underlying the beneficial (green) and deleterious (red) bell-shaped effects of glucocorticoid hormones (GCs) on stress-induced memory performance.

**Figure 5 ijms-24-04496-f005:**
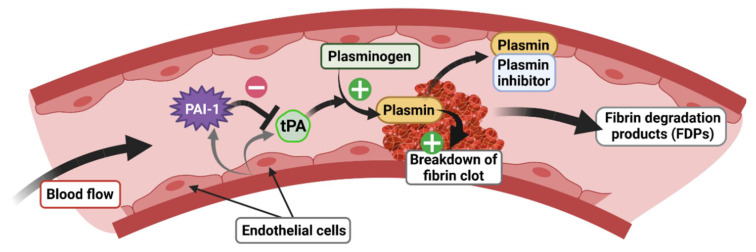
Role of the tPA/PAI-1 system in the regulation of fibrinolysis.

**Figure 6 ijms-24-04496-f006:**
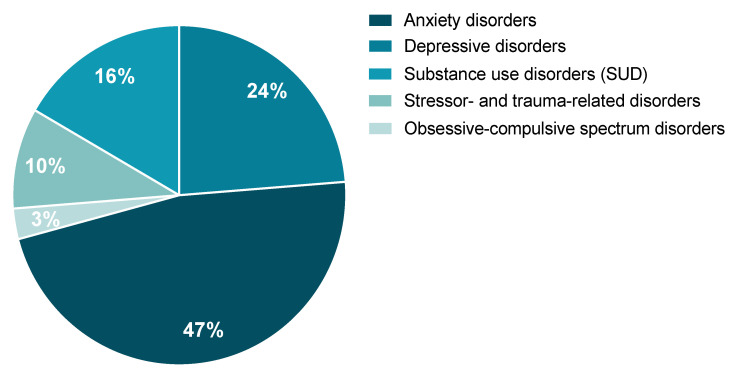
Estimated percentage of each of the five stress-related disorders of interest represented as part of a whole. Data were reanalyzed from the prevalence published in [8].

**Table 1 ijms-24-04496-t001:** Non-exhaustive list of reviews covering pathological conditions outside the CNS linked with tPA/PAI-1 balance deregulations.

tPA/PAI-1 System in	References
Cardiovascular diseases	[86,87,88]
Metabolic disorders	[89,90]
Fibrosis	[91,92]
Cancer	[93,94,95]
Inflammation and infectious diseases	[96,97,98]

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
