# Peer review of "Glucocorticoid-Responsive Tissue Plasminogen Activator (tPA) and Its Inhibitor Plasminogen Activator Inhibitor-1 (PAI-1): Relevance in Stress-Related Psychiatric Disorders"

_ijms, 2023, doi:10.3390/ijms24054496_

Round 1

Reviewer 1 Report

Mennesson and Revest addressed the major role played by tPA/PAI-1 balance modulated by GR activation in response to GC release in the context of stress-related psychiatric disorders. The authors are experts in this field. This review article is well written and can be published in the present from.

Only a few typos are detected:

Line 45; orgnanism. [11].; line 219, please specify CA1, CA2, and CA3; line 261, . Preclinical; line 346 OCD here and OCD at line 435 be deleted; line 355 ...; line 374 1,2%, 4,9%; Table, Age, months or years;

Author Response

We thank reviewer#1 for her/his positive evaluation of our study.

The typos have now been corrected in the manuscript.

Concerning OCD, we chose to differentiate the name of the class (obsessive-compulsive spectrum disorders) from the name of the disease (obsessive-compulsive disorder = OCD).

Reviewer 2 Report

I thank the authors for the opportunity to review this interesting and well-written review. The authors have prepared a comprehensive review of the field complemented by a detailed discussion of their research. I have no concerns with the content, nor the presentation or length of this material. 

Author Response

We would like to thank reviewer#2 for her/his positive evaluation of our study.

Reviewer 3 Report

This is a very well-written and comprehensive review on the topic. As a reviewer I can recommend the publication of this review paper.

However, I have two questions to the authors. 

1. Is there any gender specific effect of this imbalance in the context of PTSD? 

2. If and how juvenile trauma can contribute to such imbalance which ultimately may lead to PTSD like behavior later in life?

Author Response

We would like to thank reviewer#3 for her/his positive evaluation of our study.

To answer your questions:

1. Is there any gender specific effect of this imbalance in the context of PTSD? 

Some studies carried out in humans showed that the prevalence of stress-related pathologies is higher in females, notably for PTSD (1, 2, 3). However, to our knowledge no gender specific study has yet focused on the expression profile of the tPA/PAI-1 balance in PTSD.

  1. Yehuda, R. et al. PNat. Rev. Dis. Primer 1, 15057 (2015).
  2. Kessler, R. C et al. Arch. Gen. Psychiatry 52, 1048–1060 (1995).
  3. Breen, M. S. et al. Neuropsychopharmacol. Off. Publ. Am. Coll. Neuropsychopharmacol. 43, 469–481 (2018).

2. If and how juvenile trauma can contribute to such imbalance which ultimately may lead to PTSD like behavior later in life?

The point raised by the reviewer is extremely interesting but rather complex. Indeed, studies have shown that children subjected to traumatic events during childhood (i.e., adverse childhood experiences (ACEs)) are likely to develop behavioral and cognitive deficits later in life, which might predispose to stress-related psychiatric disorders such as PTSD and depressive disorders (1, 2, 3). However it is not clear how trauma during childhood would contribute to PTSD-like behavior later in life. Concerning tPA/PAI-1 balance, as shown in the study from Aksu et al., (2018, 4) tPA and BDNF level deregulations are observed in association with PTSD in subjects with traumatic events during childhood (sexual abuse). Unfortunately, to our knowledge no longitudinal study has yet focused on the expression profile of the tPA/PAI-1 balance in PTSD.

  1. Scheller-Gilkey et al. Early life stress and PTSD symptoms in patients with comorbid schizophrenia and substance abuse Schizophr. Res. (2004).
  2. McDonald R, Jouriles EN, Ramisetty-Mikler S, Caetano R, Green CE. Estimating the number of American children living in partner-violent families. J Fam Psychol. 2006;20(1):137–142.
  3. Pietrek C, Elbert T, Weierstall R, Muller O, Rockstroh B. Childhood adversities in relation to psychiatric disorders. Psychiatry Res. 2013;206(1):103–110.
  4. Aksu, S.; Unlu, G.; Kardesler, A.C.; Cakaloz, B.; Aybek, H. Altered Levels of Brain-Derived Neurotrophic Factor, ProBDNF and Tissue Plasminogen Activator in Children with Posttraumatic Stress Disorder. Psychiatry Res 2018, 268, 478–483, doi:10.1016/j.psychres.2018.07.013.

Reviewer 4 Report

I read the review by Marie Mennesson and Jean-Michel Revest entitled "Glucocorticoid responsive tissue plasminogen activator (tPA) and its inhibitor plasminogen activator inhibitor-1 (PAI-1): relevance in stress-related psychiatric disorders". The topic is particularly interesting and innovative.

Here are my comments

-Table 1 seems to be irrelevant to the topic, the authors should think about removing it (how were these articles selected?)

-I would ask the authors to give methodological details of how they selected the articles in Table 2.

-In Table 2, it is useful to add in the References column the name of the first author of the article.

Author Response

We would like to thank reviewer#4 for her/his positive evaluation of our study.

To answer your concerns:

-Table 1 seems to be irrelevant to the topic, the authors should think about removing it (how were these articles selected?)

The reviewer's comment is comprehensible since it is not the topic of this narrative review. However, for the sake of consistency of reading and scientific historical context, we thought it was necessary to acknowledge even briefly that numerous studies on the tPA/PAI-1 balance were first initiated outside the central nervous system (CNS). Although, we agree that Table 1 is very succinct and does not reflect the extensive literature on the tPA/PAI-1 balance outside the CNS, there are numerous studies on this topic and we could not mention them all. Thus, we selected relevant and recent reviews on the different topics and have specified in the legend that it was a non-exhaustive reference list. For these reasons, if reviewer#4 agrees, we would like to keep this table.

-I would ask the authors to give methodological details of how they selected the articles in Table 2.

We addressed this point by conducting an exhaustive search and then cross-referencing the different scientific variables relevant to the problematic (i.e., disorder, clinical trials, biological markers, etc.) using bibliographic databases mainly PubMed but also Google Scholar. Then we evaluated the relevance of the results to restore the information useful to the understanding of the problematic.

-In Table 2, it is useful to add in the References column the name of the first author of the article.

We agree with reviewer#4. This point has now been addressed in Table 2.